# Three-Dimensional Analysis of the Cranial Base Structure in Patients with Facial Asymmetry

**DOI:** 10.3390/diagnostics14010024

**Published:** 2023-12-22

**Authors:** Yuki Hayashi, Chie Tachiki, Taiki Morikawa, Yasuo Aihara, Satoru Matsunaga, Keisuke Sugahara, Akira Watanabe, Takakazu Kawamata, Yasushi Nishii

**Affiliations:** 1Department of Orthodontics, Tokyo Dental College, 2-9-18 Chiyoda-Ku, Tokyo 101-0061, Japan; yuukihayashi@tdc.ac.jp (Y.H.); morikawataiki@tdc.ac.jp (T.M.); nishii@tdc.ac.jp (Y.N.); 2Department of Neurosurgery, Tokyo Women’s Medical University, 8-1 Shinjuku-Ku, Tokyo 162-8666, Japan; aihara.yasuo@twmu.ac.jp (Y.A.); tkawamata@twmu.ac.jp (T.K.); 3Department of Anatomy, Tokyo Dental College, 2-9-18 Chiyoda-Ku, Tokyo 101-0061, Japan; matsuna@tdc.ac.jp; 4Department of Oral Pathobiological Science and Surgery, Tokyo Dental College, 2-9-18 Chiyoda-Ku, Tokyo 101-0061, Japan; ksugahara@tdc.ac.jp; 5Department of Oral & Maxillofacial Surgery, Tokyo Dental College, 2-9-18 Chiyoda-Ku, Tokyo 101-0061, Japan; akirawat@tdc.ac.jp

**Keywords:** facial asymmetry, cranial deformation, skeletal mandibular prognathism, three-dimensional (3D) deviation, computed tomography (CT)

## Abstract

Facial asymmetry is often seen in patients with skeletal mandibular prognathism and is associated with deformities in the maxillofacial and head regions. The maxillofacial deviation is three-dimensional and affects not only the lateral deviation of the mandible and midface, but also the cranium. This study conducted a three-dimensional morphological evaluation of the cranial base morphology of patients with skeletal mandibular prognathism (ANB < 0°, Wits < 0 mm) with the aim of examining the relationship between deformities of the head region and facial asymmetry. Data obtained from computed tomography conducted during the initial examination of patients with and without skeletal mandibular prognathism with facial asymmetry were used. Differences in the position of structures present in the cranial base were measured, and the association between cranial deformities and mandibular deviation was assessed. The middle cranial base area and the lateral deviation of the mandibular fossa were significantly larger in patients with facial asymmetry compared to those without facial asymmetry. In addition, a correlation between the amount of mandibular deviation and the area of the anterior cranial base was identified in patients with significant cranial deformity (*p* = 0.012). Given the identified association between the structure of the head region and facial asymmetry, further studies are needed to determine the factors implicated in the growth process.

## 1. Introduction

Jaw deformities are symptoms of abnormal direction and extent of maxillofacial growth, malocclusion and facial deformity due to the incongruity of the maxillary and mandibular arches. Among patients with jaw deformities, facial asymmetry is often caused by a three-dimensional deviation of the mandible, including deviations in rotation and tilt [1]. Facial asymmetry has also been shown to be a result of cranial deformities detected on axial cephalometric radiographs [2].

Previous reports have shown that mandibular malalignment often extends to the midface, and deformities may occur not only in the temporal bone and maxilla, but also in the cranium [3,4,5,6]. Craniofacial growth involves the cranium, midface and lower face, and the growth of these three areas is thought to be coordinated. The cranial base is classified into the anterior, middle, and posterior cranial base, and the middle cranial base includes the bone around the temporomandibular joint, whose growth is thought to likely influence that of the mandible, possibly causing its deviation and facial asymmetry. Abnormal growth of the middle cranial base could lead to deviations in the mandible and ultimately to facial asymmetry [7,8,9,10].

Patients with jaw deformities often become symptomatic during puberty when the mandible undergoes rapid growth. In 1992, the American Pediatric Society recommended that infants sleep on their backs because sleeping on their front or side is a risk factor for sudden infant death syndrome, and the incidence of cranial deformity in infants has increased significantly since then [11]. Cranial deformities are believed to occur when infants are placed in the supine position, but since these deformities are not believed to cause growth and developmental delays or functional impairment, aside from lingering aesthetic effects, they are considered acceptable [12]. However, the association between plagiocephaly and delayed intellectual and motor development has begun to attract renewed attention, and cranial deformities are now being treated in infancy [13]. It has also been speculated that cranial deformities extending into the anterior and posterior cranial regions during infancy are compensated for by the deviation of the surrounding structures [14,15]. This suggests that cranial deformities may influence the direction of subsequent growth of the middle and posterior cranial floor and of the mandible [4,5,16]. The developmental history of plagiocephaly has been the subject of much debate.

The relationship between mandibular deviation and cranial deformities in regard to jaw deformities has been evaluated by 2D analysis of cranial morphology using cephalograms [2], but data from 3D analysis of cranial morphology are scarce. In recent years, CT imaging has been used in the treatment of jaw deformities for diagnosis, orthognathic surgery simulation, and treatment evaluation. Positioning is performed by moving the jaws in all directions according to pitch, roll, and jaw parameters, and the jaws must be correctly positioned in 3D space [17,18]. Understanding distortions in three dimensions is important in surgical planning and evaluation, and the aim of this study was to understand deformations spanning the cranium in 3D. In the present study, we hypothesized that patients with facial asymmetry have cranial deformities and evaluated the 3D morphology of the cranial base to examine the relationship between deformities of the head region and facial asymmetry.

## 2. Materials and Methods

### 2.1. Research Subjects and Materials

This is a retrospective, diagnostic study which aimed at comparing 2 patient groups using cranial CT data of patients with skeletal mandibular prognathism who had visited the Tokyo Dental College Chiba Dental Center from 2010 to 2020 and underwent surgical orthodontics. Out of the 4022 patients who presented to our hospital during that period who received orthodontic treatment, there were 893 patients with jaw deformity. Skeletal mandibular prognathism was diagnosed in 769 patients, and among them, patients were randomly selected for this study. The selection criteria were based on the results of cephalometric analysis and were ANB angle less than 0° [19], Wits appraisal less than 0 mm [20,21] and menton (Me) deviation more than 4 mm in the group with facial asymmetry (asymmetry group) and less than 3 mm in the group without facial asymmetry (symmetry group). The exclusion criteria were hereditary or congenital diseases, endocrine and metabolic abnormalities, a history of severe craniomaxillofacial trauma and severe TMJ disorder. As a result, 60 patients were selected and classified into the symmetric and asymmetric groups. The characteristics of the participants are shown in Table 1.

Sample size estimation was based on our preliminary investigation. According to previous studies [3,4,5,6], this study was powered to detect differences in the area of the middle cranial base within the symmetry group and the asymmetry group, which were then compared between the two groups (*n* = 53; 75% power; 5% significance level; 1-tailed). The area differences were 1250.79 ± 868.92 mm^2^ for the symmetry group and 1735.95 ± 1222.04 mm^2^ for the asymmetry group. Considering the potential dropouts, *n* = 60 was thought to be the necessary number of patients. The calculation was carried out using software G* Power (version 3.1.9.6 for Windows; University of Düsseldorf, Düsseldorf, Germany).

Cranial morphology was measured using X-ray computed tomography (CT) images obtained before the start of the preoperative orthodontic treatment using a Somatom Plus 4 Volume Zoom^®^, Somatom Definition AS^®^ (Siemens, Erlangen, Germany) with the following settings: X-ray tube voltage, 120 kV; X-ray tube current, 117, 88 mA; slice thickness, 1.25, 1.0 mm; and slice interval, 1.0, 1.0 mm. The obtained images were output in DICOM format and measured using maxillofacial treatment simulation software (SimPlant Pro 2011; Materialise Dental, Leuven, Belgium). The same evaluator performed all 3D measurements. For each of the seven reference points comprising the reference plane, measurements were retaken 4 weeks later, and then the intraclass correlation coefficient was calculated (1, 0, 0.84).

This study was approved by the Ethics Committee of the School of Dentistry, Tokyo Dental College (approval No. 916).

### 2.2. Setting of Measurement Points, Reference Planes and Measurement Items in CT Images

The following reference planes were established on the constructed 3D images: the horizontal reference plane (X), the mid-sagittal reference plane (Y), and the frontal reference plane (Z). The horizontal reference plane was defined as the plane passing through the three points of the bilateral porions (PoR, PoL) and the left orbitale (OrL). The mid-sagittal reference plane was defined as the plane perpendicular to the horizontal reference plane that passed through the nasion (N) and the basion (Ba). The frontal reference plane was defined as the plane perpendicular to the horizontal reference plane that passed through the bilateral foramen spinosum (FsR and FsL). A plane parallel to the horizontal reference plane through the glabella was used to compare the areas of the anterior, middle and posterior cranial base (Figure 1).

A total of six reference points were defined to construct the reference plane (Figure 1 and Figure 2, Table 2).

Anatomical structures present on the left and right sides were also used as measurement points. Three points were located in the anterior cranial base, eight points in the middle cranial base, and three points in the posterior cranial base (Figure 1, Table 2).

### 2.3. Measurement and Evaluation

The distances (mm) from each measurement point to the frontal reference plane, horizontal reference plane, and mid-sagittal reference plane were measured. Differences in the left–right measurements from the deviating to the uninvolved side were established for each group. These measurements differences were compared between the facial symmetry and asymmetry groups.

Cross-sectional images were created in a plane parallel to the horizontal reference plane passing through the glabella. The cranial base was divided into three sections (anterior, middle and posterior cranial base) and into two sections (left and right) in the mid-sagittal reference plane, and the area of each section (mm^2^) was measured (Figure 2). The anterior, middle and posterior cranial base areas were established by referring to the bony prominences of the anatomical structures in the 3D images. Differences in the left–right measurements between the deviated and the uninvolved side were defined, and the left–right differences for each region were compared between the facial symmetry and asymmetry groups.

For the facial asymmetry group, cranial asymmetry (CA) and the cranial asymmetry width index (cranial vault asymmetry index, CVAI) were used to classify the degree of cranial deformity [22]. The ratio of the area of the cranial vault to the amount of mandibular deviation when the area of the non-swivel side was set to 1 was evaluated for subjects classified into the moderate and severe categories.

### 2.4. Statistical Analysis

Comparisons of the left–right differences in the anteroposterior, vertical, and lateral positions of each anatomical structure and in cranial base morphology between the symmetry and the asymmetry groups were made using the *t*-test without correspondence. In addition, correlations between the left–right difference and the ratio of cranial base area to mandibular deviation were examined using Pearson product ratio correlation analysis.

Statistical analysis was performed using SPSS (version 24.0; IBM Corporation, Armonk, NY, USA). *p* values less than 0.05 were considered statistically significant. In addition, standardized effect sizes (ESs, Cohen’s d) were calculated. The standardized effect sizes were small (0.20~0.49), medium (0.50~0.79) and large (≥0.80).

## 3. Results

The orbitales present in the frontal cranial base were at a significantly larger distance from the Z-plane in the facial asymmetry than in the facial symmetry group. Small effect sizes were obtained in the Y and Z planes (effect size, Y = 0.32, Z = 0.43). The distance from the Y-plane was greater for the apex of the mandibular condyle, the lowest point of the mandibular fossa, the pterygoid hamulus, and the articular process, which are all located in the mesocranial base. For all of them, we obtained small effect sizes (effect size, At = 0.32, Co = 0.43, Mf = 0.44, Ph = 0.45). The deepest point of the mandibular fossa showed a trend toward a greater distance from the X- and Z-planes, although this was not statistically significant. The distance from the Y-plane was greater for the mastoid on the posterior cranial floor, and that from the Z-plane was greater for the styloid process (Table 3).

The area differences for the middle cranial base were 1250.79 mm^2^ for the facial symmetry group and 1735.95 mm^2^ for the facial asymmetry group. In other words, the area difference for the cranial base was significantly larger in the facial asymmetry than in the facial symmetry group (*p* = 0.012, effect size = 0.46). The anterior and posterior cranial bases did not show significant differences (Figure 3).

After the evaluation of plagiocephaly, 10 patients were diagnosed as having moderate plagiocephaly and 2 as having severe plagiocephaly. In these subjects, the area of the cranial base in the deviated side showed a strong positive correlation with the amount of mandibular deviation (mm) in the frontal cranial base (*p* = 0.012) (Figure 4), when the area in the non-altered side was set to 1.

The area of the cranial base of the facial asymmetry group without plagiocephaly did not correlate with the amount of mandibular deviation in any cranial base region (Figure 5).

## 4. Discussion

### 4.1. Cranial Deformity in Patients with Facial Asymmetry

The results of this study indicated that in the left and right sides of the cranium, there were significant differences in the positions of the articular process, pterygoid hamulus, styloid process and mastoid between patients with facial asymmetry and those with facial symmetry. These are attachment sites for masticatory and perioral muscles and are therefore closely related to functional aspects such as mandibular movement [15,23]. Differences in the anatomical position of the structures may result in asymmetrical functional movements and contribute to the growth of facial asymmetry [24,25,26].

Patients with facial asymmetry showed significant differences in left–right measurements in the area of the middle cranial base. A significant difference in the lateral left–right measurements was also observed for the mandibular fossa. These results suggest that patients with facial asymmetry exhibit changes in mandibular fossa position and middle cranial base morphology. Previous studies reported that 3D left–right differences in the mandibular fossa are associated with facial and mandibular asymmetry [7,8,9,10]. The mandibular fossa resides in the temporal bone and constitutes the TMJ [27], and its alteration leads to deformities in the middle cranial base, causing asymmetry of the mandible [28]. In other words, it can be inferred from the present study that facial asymmetry due to deformation of the midcranial floor leads to mandibular asymmetry based on the deviation of the mandibular fossa [26]. Because mandibular growth occurs primarily in the condylar region [29], mandibular condyle deviation affects the position of the mandibular head [30], causing lateral deviation of the mandible. In short, the secondary shape distortion of the growing infant skull caused by the supine position could cause cranial deformities and mandibular morphology deformities [24,28]. The correction of cranial deformities should be considered as part of orthotic therapy for patients with plagiocephaly in order to reduce facial asymmetry and its associated developmental consequences as early as possible. Therefore, further analysis of the relationship between the amount of cranial deformation and the direction of the deformation axis during growth and development, and of the deformation of the mandible itself, is needed.

### 4.2. Relationship between Cranial Deformity and Mandibular Deviation in Patients with Facial Asymmetry with Plagiocephaly

Plagiocephaly is a condition in which CA, particularly unilateral flattening of the frontal and posterior cranial portions, occurs due to various factors [14,26]. There is no objective way to assess CA in adults. On the other hand, in infants, CA and the CVAI are commonly used to evaluate the severity of cranial deformity [22]. Therefore, we focused on CA and the CVAI in this study. Kwon et al. [5] reported that patients with skeletal mandibular prognathism with facial asymmetry have mild plagiocephaly. In the present study, one-fifth of the patients in the facial asymmetry group had moderate or severe deformational plagiocephaly, and the amount of mandibular deviation in patients with deformational plagiocephaly showed a strong positive correlation with the area of the anterior cranial base (*p* = 0.012). Because the growth of the anterior cranial base is almost complete at the age of 5 years [31], the deviation of the anterior cranial base is thought to influence the direction and amount of subsequent mandibular growth and deviation.

For the posterior cranial base, the difference in the cranial base area was greater in patients with facial symmetry than in those with facial asymmetry. No correlation was found between the amount of mandibular deviation and the areas of the left and right cranial base in patients with facial asymmetry. Although it was suggested that CA occurs during infancy [4], no mention has been made of cranial base asymmetry after growth. This is likely because the occipital area continues to grow until late adolescence [7,29]. On the contrary, it has been reported that facial asymmetry is observed in 10% of children at the age of 6 years and that it is sometimes no longer observed around the age of 16 years [32]. The results of the present and previous studies suggest that asymmetry may remain in the anterior and middle cranial bases, where growth ends early, while the posterior cranial base may grow to compensate for the asymmetry. In particular, the significant correlation found between the area of the middle cranial fossa (Figure 2), which contains the pyramidal bone where the auricle (external auditory canal, internal auditory canal and tricuspid) and TMJ are anatomically located, and the facial bone asymmetry complication rate is considered a valuable finding. However, because some studies reported an association between mandibular deviation and cranial flattening on the uninvolved side [33], it is necessary to examine relevant factors further in the future.

### 4.3. Research Methods

In clinical diagnosis, facial asymmetry is often analyzed using 2D radiographs. In recent years, CT images have been used in the analysis of jaw deformities for diagnosis, simulation of orthognathic surgery and treatment evaluation. Actual-size 3D measurements can be taken regardless of the head position at the time of imaging, and the area of 3D structures can also be measured. However, there is no definition of the reference plane on a 3D image. The external auditory canal and foramen magnum are considered to have a stable shape and are therefore considered to be relatively unaffected by craniomaxillofacial deformation [34]. In reference to previous 3D measurement studies, the Frankfurt horizontal plane passing through the bilateral porions and left lateral orbitale was used as the horizontal reference plane in the present study [5]. Regarding the median reference plane, the nasion and basion were chosen as the median reference plane because they are highly consistent structures in 3D [35] and are often used as measurement items in the mid-sagittal plane [36].

Facial asymmetry is more common in patients with skeletal mandibular prognathism than in those with skeletal maxillary prognathism [1,3,37,38,39]. Skeletal asymmetry has been reported to increase as the craniofacial bones move from upper to lower positions and from posterior to anterior positions, even in subjects with normal occlusion. The most pronounced growth is seen after the peak growth period of puberty [37]. The mandible grows particularly rapidly during this period, and this is most noticeable in cases of skeletal mandibular prognathism [40]. In particular, facial asymmetry is often associated with lateral mandibular deviation, which is an anteroposterior as well as a horizontal axial abnormality [1,7,8]. According to the criteria of the American Association of Oral and Maxillofacial Surgeons, a mandibular midline deviation of 3 mm or more is considered facial asymmetry [41]. Haraguchi et al. [1] considered the critical deviation of approximately 4 mm from the median sagittal plane to distinguish between facial symmetry and asymmetry, and Masuoka et al. [42] found that the average Me distance from the median sagittal plane on cephalograms of patients deemed by orthodontists to require orthognathic surgery for facial asymmetry was 3.45–4.89 mm. In the present study, a Me distance from the median sagittal plane of more than 4 mm on CT was considered for patient classification; therefore, the patients whose Me deviated more than 4 mm laterally from the median sagittal plane on CT were included in the facial asymmetry group. In reference to Japanese data [10,38,42] and to measurements indicating normality in 3D cephalometrics [43], the boundary of the facial symmetry group in this study was set to 3 mm or less.

### 4.4. Clinical Application

The results of this study indicated that cases of midbasal cranial deformity were significantly more frequent in patients with facial asymmetry. In patients with facial asymmetry with minimal cranial deformity, no correlation was found between the amount of mandibular deviation and the degree of cranial deformity. However, a significant correlation was found between frontal cranial floor deformity and the amount of mandibular deviation in patients with facial asymmetry and plagiocephaly. Because the frontal skull base grows vigorously during infancy, we suggest the possibility that skull base deformity during infancy may be highly related to mandibular deviation and facial asymmetry after growth. Therefore, it is suggested that treating cranial deformity in infancy may promote the normal development of cranial morphology, reduce developmental consequences as early as possible and prevent facial asymmetry. In the future, we believe that a prospective study in collaboration with researchers in other fields, including a randomized controlled trial, could provide highly accurate recommendations for clinical practice in the field of oral and maxillofacial surgery.

## 5. Conclusions

In the present study, a 3D analysis of cranial base morphology in patients with facial asymmetry showed that the area of the middle cranial base and the position of the mandibular condyle were significantly related to facial asymmetry. In patients with significant cranial deformity, a moderate correlation was found between the amount of mandibular deviation and the area of the anterior cranial base.

## Figures and Tables

**Figure 1 diagnostics-14-00024-f001:**
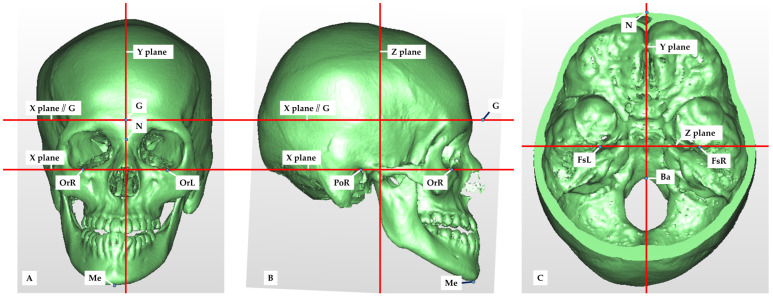
Reference planes and reference points. (**A**) Frontal surface 3D image reconstructed by medical CT data. N: nasion, OrR, OrL: orbitale, G: glabella (a point on soft tissue), Me: menton, X-plane: horizontal reference plane, Y-plane: mid-sagittal plane, X-plane//G: plane parallel to the horizontal reference plane through the glabella. (**B**) Lateral surface. PoR: porion, Z-plane: frontal reference plane. (**C**) Internal surface of the cranial base. Ba: basion, FsR, FsL: foramen spinosum.

**Figure 2 diagnostics-14-00024-f002:**
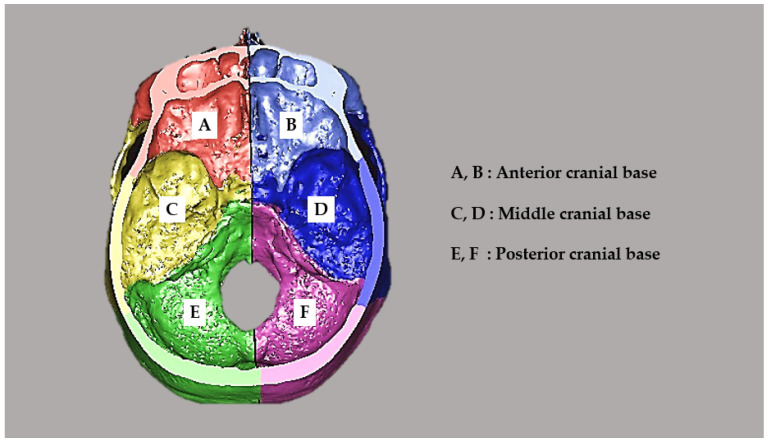
Segmentation of the cranial base. The area was cut by a plane parallel to the horizontal reference plane through the glabella and measured from that plane to the cranial base. Each section of the cranial base was divided into a left–right pair by the mid-sagittal reference plane.

**Figure 3 diagnostics-14-00024-f003:**
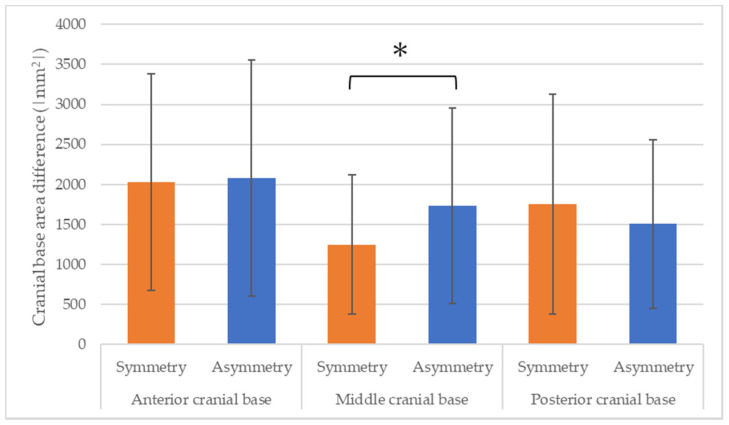
Left–right differences in morphology for each cranial base area (* *p* < 0.05).

**Figure 4 diagnostics-14-00024-f004:**
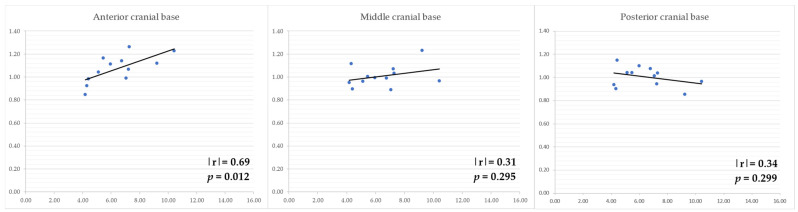
Correlation between the area of each cranial base region and mandibular deviation in the facial asymmetry group with plagiocephaly. Vertical axis: the amount of mandibular deviation (mm) in each cranial base region when the area of the non-altered side was set to 1. Horizontal axis: the amount of menton deviation (mm).

**Figure 5 diagnostics-14-00024-f005:**
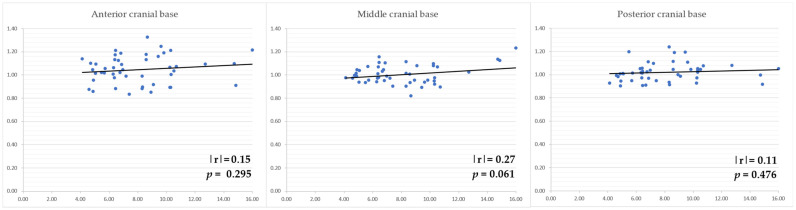
Correlation between the approximate area of each cranial base region and the amount of mandibular deviation in the facial asymmetry group without plagiocephaly. Vertical axis: the amount of mandibular deviation (mm) in each cranial base region when the area of the non-altered side was set to 1. Horizontal axis: the amount of menton deviation (mm).

**Table 1 diagnostics-14-00024-t001:** Patients’ characteristics.

	Symmetry (*n* = 60)	Asymmetry (*n* = 60)
Sex		
Male	20	16
Female	40	44
Age (y)		
Mean	20.9	24.4
Range	14.9–49.9	14.2–49.4
Measurement values		
Mean ANB (°)	−2.85	−3.34
Mean Wits appraisal (mm)	−11.70	−10.22
Difference in position of Me (mm)	1.75	7.85
Range in position of Me (mm)	0.14–2.59	4.19–16.29

**Table 2 diagnostics-14-00024-t002:** Definition of the reference and measurement points.

	Abbreviation	Explanation
*Reference point*		
Glabella	G	Most prominent point between the eyebrows
Nasion	Na	Most anterior point of the frontal nasal suture
Orbitale	Or	Lowest point of the orbital bone margin
Porion	Po	Sublingual neural tube opening
Foramen spinosum	Fs	Opening of the foramen spinosum
Basion	Ba	Lowest point on the anterior margin of the foramen magnum occipitalis
Menton	Me	Lowest point of the mandibular symphysis
*Measurement point*		
Anterior cranial base		
Orbitale	Or	Lowest point on the orbital bony margin
Frontozygomatic suture	Fz	Most medial point of the zygomatico-frontal suture
Anterior clinoid process	Acl	Last point of the anterior process of the anterior floor
Middle cranial base		
Posterior clinoid process	Pcl	Most lateral point of the posterior process of the floor
Sphenoid lesser wing	Sl	Most centrifugal point of the lesser wing of the sphenoid
Foramen rotundum	Fr	Foramen magnum opening
Foramen spinosum	Fs	Foramen spinosum opening
Articular process	At	Lowest point of the articular process
Condyle	Co	Uppermost point of the condyle
Mandibular fossa	Mf	Deepest point of the mandibular fossa
Pterygoid hamulus	Ph	Most posterior point of the pterygoid hook
Posterior cranial base		
Styloid process	Sty	Lowermost point of the stromal process
Mastoid process	M	Lowest point of the mastoid process
Hypoglossal canal	Hyp	Sublingual neural tube opening

The anatomical structures to be measured were selected as pairs on the left and right sides.

**Table 3 diagnostics-14-00024-t003:** Measurements of 3D distances for each anatomical structure (|mm|).

	X					Y					Z				
	Symmetry		Asymmetry			Symmetry		Asymmetry			Symmetry		Asymmetry		
	Avg.	SD	Avg.	SD	*p*	Avg.	SD	Avg.	SD	*p*	Avg.	SD	Avg.	SD	*p*
*Anterior cranial base*															
Or	0.93	0.71	0.92	0.72	NS	1.48	0.91	1.86	1.39	NS	1.48	1.21	2.23	2.13	0.02 *
Fz	1.52	1.05	1.52	1.10	NS	0.69	0.65	0.87	0.84	NS	2.07	1.55	2.80	2.41	NS
Acl	1.00	1.14	0.90	0.66	NS	1.21	0.86	1.23	0.91	NS	1.10	0.93	1.00	0.84	NS
*Middle cranial base*															
Pcl	1.71	2.28	1.38	0.96	NS	1.28	0.92	1.36	1.19	NS	1.01	1.18	0.92	0.65	NS
Sl	2.29	1.44	2.11	1.70	NS	2.60	2.12	2.65	2.35	NS	2.17	1.75	2.34	1.85	NS
Fr	1.40	1.37	1.29	1.09	NS	1.36	1.13	1.71	1.42	NS	1.65	1.11	1.84	1.81	NS
Fs	0.00	0.00	0.00	0.00	NS	2.11	1.49	2.23	1.96	NS	2.14	1.59	2.34	1.83	NS
At	1.35	0.97	1.20	1.04	NS	1.42	0.91	1.88	1.22	0.03 *	3.07	2.19	2.99	2.45	NS
Co	0.70	0.70	0.88	0.71	NS	1.60	1.41	2.20	1.62	0.04 *	1.62	1.31	1.89	1.57	NS
Mf	0.79	0.73	0.75	0.66	NS	1.30	1.18	1.97	1.79	0.02 *	1.68	1.36	1.79	1.27	NS
Ph	0.89	0.72	0.97	0.81	NS	1.75	1.18	2.45	1.83	0.02 *	1.19	0.91	1.28	1.09	NS
*Posterior cranial base*															
Sty	2.77	3.11	3.17	3.37	NS	2.34	1.80	3.16	2.69	NS	1.67	1.51	2.61	2.96	0.04 *
M	1.24	0.99	1.63	1.74	NS	1.63	1.61	2.73	3.36	0.03 *	2.28	1.96	2.99	3.21	NS
Hyp	1.22	1.12	1.25	1.90	NS	1.56	2.19	1.42	1.32	NS	1.04	0.75	1.07	1.01	NS

The vertical distance from each plane (X, Y, Z) was measured (* *p* < 0.05). Average; Avg., standard deviation; SD, *p*-value; *p*, not significant; NS.

## Data Availability

The data presented in this study are available upon request from the corresponding author.

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
