# Peer review of "Three-Dimensional Analysis of the Cranial Base Structure in Patients with Facial Asymmetry"

_diagnostics, 2023, doi:10.3390/diagnostics14010024_

Round 1
Reviewer 1 Report
Comments and Suggestions for Authors
This is interesting paper which used CT images to objective the cranial anatomy in relation the symmetry. There are some points which should be taken into consideration by the authors:
- The paper is in present form not fully and correct formatted. Please revise carefully. In detail, legends are not located on the same page of the table/ figure.
- Table 2 is printed over two pages.
- Table 3: the width of all columns can be reduced. Is the font size correct?
- The abstract is free of any data! – Please revise carefully and include relevant numeric information from your study.
- Line 79: You have performed a retrospective, diagnostic study which aimed at comparing two patient group using cranial CT data.
- Line 89: please change .05 into “0.05”
- Chapter 2.3 – Is it necessary to have subheadings for this small chapter?
Please see above comments.
Author Response
Ref.: diagnostics-2737404
Three-Dimensional Analysis of the Cranial Base Structure in Patients
with Facial Asymmetry
December 17, 2023
Dear Reviwer 1,
Thank you for your email of November 31, 2023, regarding our manuscript,
“Three-Dimensional Analysis of the Cranial Base Structure in Patients
with Facial Asymmetry”, and the valuable comments of the three reviewers. I attach here our revised manuscript, as well as a point-by-point response to the reviewers’ comments.
We feel that the revised manuscript is a suitable response to the comments, and is significantly improved over the initial submission. We trust that it is now suitable for publication in the Diagnosis.
Thank you in advance for your kind consideration of this paper.
Sincerely yours,
Yuki Hayashi
Resident,
Department of Orthodontics, Tokyo Dental College,
2-9-18, Kanda-misaki-cho, Chiyoda-ku, Tokyo, 101-0061, Japan.
yuukihayashi@tdc.ac.jp
For research article
Response to Reviewer 1 Comments
|
||
1. Summary |
|
|
Thank you very much for spending your time for the review of our paper, and for very valuable comments to improve the manuscript. The authors have revised the manuscript according to all the comments and questions as follows. |
||
|
||
2. Questions for General Evaluation |
Reviewer’s Evaluation |
Response and Revisions |
Does the introduction provide sufficient background and include all relevant references? |
Yes |
|
Are all the cited references relevant to the research? |
Yes |
|
Is the research design appropriate? |
Yes |
|
Are the methods adequately described? |
Yes |
|
Are the results clearly presented? |
Yes |
|
Are the conclusions supported by the results? |
Yes |
|
|
|
|
3. Point-by-point response to Comments and Suggestions for Authors |
||
Comments 1: The paper is in present form not fully and correct formatted. Please revise carefully. In detail, legends are not located on the same page of the table/ figure. Comments 2: Table 2 is printed over two pages. Comments 3: Table 3: the width of all columns can be reduced. Is the font size correct? |
||
Response 1-3: We wish to thank the reviewer for this comment. In accordance with Reviewer 1’s comment, we have changed the format of table 2,3, figure 1-5, and the font size of table 3. |
||
Comments 4: The abstract is free of any data! – Please revise carefully and include relevant numeric information from your study. |
||
Response 4: We thank the Reviewer for this pertinent comment. In accordance with the Reviewer’s comment, we have added some data in this study to Abstract. |
||
Comments 5: Line 79: You have performed a retrospective, diagnostic study which aimed at comparing two patient group using cranial CT data. |
||
Response 2: We thank the Reviewer for this pertinent comment. In accordance with the Reviewer’s comment, I changed the wording of lines 76-79. Line 76-79: The subject is this retrospective, diagnostic study which aimed at comparing 2 pa-tient groups using cranial CT data of patients with skeletal mandibular prognathism who had visited the Tokyo Dental College Chiba Dental Center from 2010–2020 and underwent surgical orthodontics. |
||
Comments 6: Line 89: please change .05 into “0.05”. |
||
Response 6: In accordance with the reviewer’s comment, we have corrected the misspelling. |
||
Comments 7: Chapter 2.3 – Is it necessary to have subheadings for this small chapter? |
||
Response 7: We appreciate the reviewer's comment. In accordance with the Reviewer’s comment, We have removed small chapters of Chapter 2,3. |
||
|
||
4. Response to Comments on the Quality of English Language |
||
Point 1: Minor editing of English language required. |
||
Response 1: We thank the reviewers for their appropriate comments. The manuscript was proofread in English prior to submission. The certificate is attached for your review. |
||
|
||
5. Additional clarifications |
||
|
Reviewer 2 Report
Comments and Suggestions for Authors
1. Numbering of the pages is not running correctly
2. The authors state in the Introduction that plagiocephaly may be related to intellectual or motor delay. Hover, there are studies with non-syndromatic plagiocephaly patients expansion cranioplasty and it seems that surgery seem not to have any effect on the child development and only minor differences may be discovered while studying the brain waves postoperatively. This must be discussed at least, since in this present form as written the topic stays hanging withouth conclusion.
3. What was the total population of the patients where by excluding the study population of 60 + 60 was generated? Was it just random 60 symmetric patients? Was it just the most worst 60 asymmetric patients? It must be clear how the study groups formed.
4. Would be interesting to see if orbital volumes Left vs Right differ in symmetry or non-symmetry groups
The topic is signifficant since 3D VSP is rapidly becoming a gold standard in orthognatghic surgery and scull position is important pre planning procedure. Overall, nicely written paper and should be considered to be published with minor modifications.
Author Response
Ref.: diagnostics-2737404
Three-Dimensional Analysis of the Cranial Base Structure in Patients
with Facial Asymmetry
December 17, 2023
Dear Reviwer 2 ,
Thank you for your email of November 31, 2023, regarding our manuscript,
“Three-Dimensional Analysis of the Cranial Base Structure in Patients
with Facial Asymmetry”, and the valuable comments of the three reviewers. I attach here our revised manuscript, as well as a point-by-point response to the reviewers’ comments.
We feel that the revised manuscript is a suitable response to the comments, and is significantly improved over the initial submission. We trust that it is now suitable for publication in the Diagnosis.
Thank you in advance for your kind consideration of this paper.
Sincerely yours,
Yuki Hayashi
Resident,
Department of Orthodontics, Tokyo Dental College,
2-9-18, Kanda-misaki-cho, Chiyoda-ku, Tokyo, 101-0061, Japan.
yuukihayashi@tdc.ac.jp
For research article
Response to Reviewer 2 Comments
|
||
1. Summary |
|
|
Thank you for your recognition of this study of three-dimensional analysis on patients with jaw deformities. I appriciate for spending your time for the review of our paper, and for very valuable comments to improve the manuscript. The authors have revised the manuscript according to all the comments and questions as follows. |
||
|
||
2. Questions for General Evaluation |
Reviewer’s Evaluation |
Response and Revisions |
Does the introduction provide sufficient background and include all relevant references? |
Can be improved |
|
Are all the cited references relevant to the research? |
Yes |
|
Is the research design appropriate? |
Yes |
|
Are the methods adequately described? |
Yes |
|
Are the results clearly presented? |
Yes |
|
Are the conclusions supported by the results? |
Yes |
|
|
|
|
3. Point-by-point response to Comments and Suggestions for Authors |
||
Comments 1: Numbering of the pages is not running correctly. |
||
Response 1: We wish to thank the reviewer for this comment. In accordance with Reviewer 2’s comment, we have changed the format of numbering of page. |
||
Comments 2: The authors state in the Introduction that plagiocephaly may be related to intellectual or motor delay. Hover, there are studies with non-syndromatic plagiocephaly patients expansion cranioplasty and it seems that surgery seem not to have any effect on the child development and only minor differences may be discovered while studying the brain waves postoperatively. This must be discussed at least, since in this present form as written the topic stays hanging without conclusion. |
||
Response 2: We thank the Reviewer for this pertinent comment. We have added reference. The reference suggests that “back to sleep” causes plagiocephaly and delayed gross motor development. |
||
Comments 3: What was the total population of the patients where by excluding the study population of 60 + 60 was generated? Was it just random 60 symmetric patients? Was it just the most worst 60 asymmetric patients? It must be clear how the study groups formed. |
||
Response 2: We thank the Reviewer for this pertinent comment. The total population is 60+60 patients diagnosed with jaw deformity who came to the clinic between 2010 and 2020 and were randomly selected under these condition (facial symmetry/facial asymmetry). In accordance with the Reviewer’s comment, we have added the information about patients in this study to Material and Method. Line 86-89: Out of the 4022 patients who presented to our hospital during that period diagnosed with orthodontic treatment, there were 893 patients with jaw deformity. Skeletal mandibular prognathism was diagnosed in 769 patients, and among them, patients were randomly selected under following conditions. |
||
4. Response to Comments on the Quality of English Language |
||
Point : I am not qualified to assess the quality of English in this paper. |
||
|
||
5. Additional clarifications |
||
Comments : Would be interesting to see if orbital volumes Left vs Right differ in symmetry or non-symmetry groups The topic is signifficant since 3D VSP is rapidly becoming a gold standard in orthognatghic surgery and scull position is important pre planning procedure. Overall, nicely written paper and should be considered to be published with minor modifications. |
||
Response : Thank you for your recognition of this study of three-dimensional analysis on patients with jaw deformities. We have revised according to all of your comments as in the followings. |
Reviewer 3 Report
Comments and Suggestions for Authors
Dear authors, thanks for your kind submission and for the possibility to review it for you, below you will find my comments and suggestions to increase the quality of your manuscript,
Being involved in three dimensional analysis personally, I thank the authors for their extensive work and I hope to bring useful comments.
Introduction
Page 2 line 51 there is a typo: "mandi-ble"
In general the introduction focuses too much in establishing possible correction on the ethiology of the asymmetry, but this is not demonstrated and could lure the audience to speculations, I suggest to add updated literature and support the analysis of craniofacial anormalities with the most recent studies such as the methods used and the differences between two dimensional and three dimensional results achieved. (for example https://doi.org/10.3390/bioengineering9050216)
Materials and methods:
Please give more informations about the preliminary study mentioned in the sample size. What datas was extracted? how many cases? what numeric values did you use?
2.1
"menton (Me) deviation more than 4 mm as the group with facial asymmetry 80 (asymmetry group), and Me deviation less than 3 mm as the group without facial 81 asymmetry (symmetry group)"
Why did you use more than 4 mm? And how was calculated this deviation? Also less than 3 mm is a number that should be justified, a discrepancy between the two groups of just 1 mm maybe would not be enough?
2.2 please specify if current planes have been described previously or not, and if the methodology is new some more specifications should be given on the reliability
Also the planes pass through points but how was the horizontal plan oriented?? This is very important and was not described, three points minimum should be used to define a plane
2.3 should be better explained, how were the distances obtained?
"The left– 137 right difference in measurements was then compared between the facial symmetry and 138 asymmetry groups as the difference in measurements from the deviating to the non- 139 evolving side." this sentence is not clear to me
Results : The statistical analysis is adequately described, but it would be beneficial to include effect sizes along with p-values to assess the clinical significance of the findings.
The discussion provides a good interpretation of the results. However, the clinical implications of the findings could be elaborated upon. How might the observed cranial deformities influence treatment decisions or outcomes in patients with facial asymmetry?
Limitations and aknowledgments could be added at the end of the manuscript
References are very old, there is no need to cite so many manuscripts from the past, while some are necessary the authors should address some efforts to select the most recent updates on the topic
Author Response
Ref.: diagnostics-2737404
Three-Dimensional Analysis of the Cranial Base Structure in Patients
with Facial Asymmetry
December 17, 2023
Dear Reviwer 3,
Thank you for your email of November 31, 2023, regarding our manuscript,
“Three-Dimensional Analysis of the Cranial Base Structure in Patients
with Facial Asymmetry”, and the valuable comments of the three reviewers. I attach here our revised manuscript, as well as a point-by-point response to the reviewers’ comments.
We feel that the revised manuscript is a suitable response to the comments, and is significantly improved over the initial submission. We trust that it is now suitable for publication in the Diagnosis.
Thank you in advance for your kind consideration of this paper.
Sincerely yours,
Yuki Hayashi
Resident,
Department of Orthodontics, Tokyo Dental College,
2-9-18, Kanda-misaki-cho, Chiyoda-ku, Tokyo, 101-0061, Japan.
yuukihayashi@tdc.ac.jp
For research article
Response to Reviewer 3 Comments
|
||
1. Summary |
|
|
Thank you very much for spending your time for the review of our paper, and for very valuable comments to improve the manuscript. The authors have revised the manuscript according to all the comments and questions as follows. |
||
|
||
2. Questions for General Evaluation |
Reviewer’s Evaluation |
Response and Revisions |
Does the introduction provide sufficient background and include all relevant references? |
Can be improved |
|
Are all the cited references relevant to the research? |
Must be improved |
|
Is the research design appropriate? |
Yes |
|
Are the methods adequately described? |
Must be improved |
|
Are the results clearly presented? |
Yes |
|
Are the conclusions supported by the results? |
Can be improved |
|
|
|
|
3. Point-by-point response to Comments and Suggestions for Authors |
||
Comments 1: Page 2 line 51 there is a typo: "mandi-ble" |
||
Response 1: In accordance with the reviewer’s comment, we have corrected the missspelling. |
||
Comments 2: In general the introduction focuses too much in establishing possible correction on the ethiology of the asymmetry, but this is not demonstrated and could lure the audience to speculations, I suggest to add updated literature and support the analysis of craniofacial anormalities with the most recent studies such as the methods used and the differences between two dimensional and three dimensional results achieved. (for example https://doi.org/10.3390/bioengineering9050216) |
||
Response 2: We thank the reviewer for this comment. We have added to introduction about 3D results. “In recent years, CT imaging has been used in the treatment of jaw deformities for di-agnosis, orthognathic surgery simulation, and treatment evaluation. Positioning is done by moving the jaws in all directions according to pitch, roll, and jaw parameters, and the jaws must be correctly positioned in 3D space [17,18]. Understanding distortions in three dimensions is important in surgical planning and evaluation, and the aim of this study was to understand deformations spanning the cranium in 3D.” |
||
Comments 3: Please give more informations about the preliminary study mentioned in the sample size. What datas was extracted? how many cases? what numeric values did you use? |
||
Response 3: We thank the reviewer for this comment. In this study, we have calculated the sample size from the preliminary study, so we consider that as evidence. In accordance with the Reviewer’s comment, we have added the information about sample size in this study to Material and Method. Line 95-102: Sample size estimation was based on our preliminary investigation. According to the previous studies [3-6], This study was powered to detect on the area differences of the middle cranial base between the facial symmetry group and symmetry group ( n = 53; 75% power; 5% significance level; 1-tailed). The area differences were 1250.79±868.92 mm2 for the facial symmetry and 1735.95±1222.04 mm2 for the facial asymmetry. Considering the potential number of dropouts, n = 60 was thought to be necessary. The calculation was carried out using software G* Power (version 3.1.9.6 for Windows; University of Düsseldorf, Düsseldorf, Germany). |
||
Comments 4: Line 80,81: "menton (Me) deviation more than 4 mm as the group with facial asymmetry (asymmetry group), and Me deviation less than 3 mm as the group without facial asymmetry (symmetry group)" Why did you use more than 4 mm? And how was calculated this deviation? Also less than 3 mm is a number that should be justified, a discrepancy between the two groups of just 1 mm maybe would not be enough?. |
||
Response 4: We thank the reviewer for this comment. It may be very confusing, but we based our boundaries on Japanese data(e.g. Refs. 48,52).Also, in reference to previous studies(Refs. 2), the boundary of the facial symmetry group was defined as 3 mm. |
||
Comments 5: Please specify if current planes have been described previously or not, and if the methodology is new some more specifications should be given on the reliability. Also the planes pass through points but how was the horizontal plan oriented?? This is very important and was not described, three points minimum should be used to define a plane. |
||
Response 5: We thank the reviewer for this comment. We refer to previous studies(Refs. 45) and show the horizontal reference plane as follows. Line 117,118: The horizontal reference plane was defined as the plane passing through the three points of the bilateral porions (PoR, PoL) and the left orbitale (OrL). |
||
Comments 6: Should be better explained, how were the distances obtained? Line 137-139: "The left–right difference in measurements was then compared between the facial symmetry and asymmetry groups as the difference in measurements from the deviating to the nonevolving side." This sentence is not clear to me, |
||
Response 6: We thank the reviewer for this comment. In accordance with the Reviewer’s comment, I changed the wording of lines 134-138. Line 134-138: The left–right difference in measurements was then compared between the facial sym-metry and asymmetry groups as the difference in measurements from the deviating to the non-evolving side. These measurements difference was compared between the facial symmetry and facial asymmetry groups. |
||
Comments 7: The statistical analysis is adequately described, but it would be beneficial to include effect sizes along with p-values to assess the clinical significance of the findings. |
||
Response 7: We thank the reviewer for this comment. In accordance with the Reviewer’s comment, we conducted further studies using effect sizes.Therefore, I added a note to the Result. |
||
Comments 8: The discussion provides a good interpretation of the results. However, the clinical implications of the findings could be elaborated upon. How might the observed cranial deformities influence treatment decisions or outcomes in patients with facial asymmetry? |
||
Response 8: We thank the reviewer for this comment. In accordance with the Reviewer’s comment, we have added the discussion in the text. we have added the clinical implications in this study to Discussion. Line 326-328: Therefore, it was suggested that treating cranial deformity in infancy may promote normal development of cranial morphology, reduce developmental consequences as early as possible, and prevent facial asymmetry. |
||
Comments 9: References are very old, there is no need to cite so many manuscripts from the past, while some are necessary the authors should address some efforts to select the most recent updates on the topic. |
||
Response 9: In accordance with the Reviewers’s insightful comments, we have added new references that are relatively new. |
||
|
||
4. Response to Comments on the Quality of English Language |
||
Point 1: English language fine. No issues detected. |
||
|
||
5. Additional clarifications |
||
|
Round 2
Reviewer 3 Report
Comments and Suggestions for Authors
Dear authors, thanks for your kind revision, I appreciated the efforts from the authors and the precise responses, I am thankful for following my indications and I hope the soundness of your research increased,
I think some more revision is still needed in the bibliography, I noticed that four manuscript have been added, but this is a low percentage out of 52, and also two of the manuscripts added are more than 10 years old, therefore I invite the authors to check again all the outdated references (more than 10 years old), and old literature (20 + years) and especially the historical manuscripts (30 + years!) and question if they are really necessary for a scientific manuscript. Also it would be nice to check the symmetry normality in different ethnic groups (doi.org/10.3390/bioengineering9050216) apart from this I have no more suggestions, thanks.
Author Response
Please see the attachment.
Ref.: diagnostics-2737404
Three-Dimensional Analysis of the Cranial Base Structure in Patients
with Facial Asymmetry
December 20, 2023
Dear Reviwer 3,
Thank you for your email of December 18, 2023, regarding our manuscript,
“Three-Dimensional Analysis of the Cranial Base Structure in Patients
with Facial Asymmetry”, and the valuable comments of the three reviewers. I attach here our revised manuscript, as well as a point-by-point response to the reviewers’ comments.
We feel that the revised manuscript is a suitable response to the comments, and is significantly improved over the initial submission. We trust that it is now suitable for publication in the Diagnosis.
Thank you in advance for your kind consideration of this paper.
Sincerely yours,
Yuki Hayashi
Resident,
Department of Orthodontics, Tokyo Dental College,
2-9-18, Kanda-misaki-cho, Chiyoda-ku, Tokyo, 101-0061, Japan.
yuukihayashi@tdc.ac.jp